# Dual Application: p-CuS/n-ZnS Nanocomposite Construction for High-Efficiency Colorimetric Determination and Photocatalytic Degradation of Tetracycline in Water

**DOI:** 10.3390/nano12234123

**Published:** 2022-11-22

**Authors:** Li Zhang, Linhong Ge, Lamei Deng, Xinman Tu

**Affiliations:** 1Key Laboratory of Jiangxi Province for Persistent Pollutants Control and Resources Recycle, Nanchang Hangkong University, Nanchang 330063, China; 2National-Local Joint Engineering Research Center of Heavy Metals Pollutants Control and Resource Utilization, Nanchang Hangkong University, Nanchang 330063, China

**Keywords:** p–n heterojunction, tetracycline, colorimetric determination, photocatalysis, degradation

## Abstract

Herein, CuS was incorporated with ZnS to form a novel nanocomposite via cation exchange, and the product was then employed for dual application of the colorimetric determination and photocatalytic degradation of tetracycline (TC) in water. The formed p–n heterojunction provided an improved gap width and electron mobility, which could rapidly catalyze H_2_O_2_ to produce plenty of •OH, supporting a color conversion with TMB. Meanwhile, the addition of TC could lead to the further enhancement in colorimetric signal, and the distinction level was sensitive to the target amount. Additionally, under light conditions, the p-CuS/n-ZnS could produce •O_2_^−^, •OH, and h^+^ through photocatalysis, and these ions could degrade the TC via oxidation. In the colorimetric determination of TC, the signal responses were obtained within 10 min, and the detection limit was 20.94 nM. The recovery rates were 99% and 106% for the water samples from Ganjiang river. In the photocatalytic degradation, the TC was degraded by 91% within 120 min, which was threefold that of ZnS. Meanwhile, the morphology feature of the p-CuS/n-ZnS remained after multiple uses, suggesting a favorable material stability. This strategy provides application prospects for the monitoring and control of antibiotics in water.

## 1. Introduction

Tetracycline (TC) is commonly recognized as an efficient antibiotic to inhibit Gram-positive or -negative bacteria [1,2]. However, its abuse in medicine, agriculture, and livestock results in a contamination of water and the pollutant entering the circulatory system, further causing a menace to human health [3,4]. Therefore, it is of great significance to establish a way of monitoring and controlling TC. Recently, the main detection technologies for TC have included enzyme-linked immunosorbent assay, liquid chromatography-mass spectrometry, molecular imprinting, and electrochemical sensing [5,6,7,8]. These methods are sensitive and accurate, but require credible instruments and professional operations, which are limited in rapid detection. Moreover, these strategies have been little reported as being used for pollutant degradation. The challenges of monitoring and control for TC remain.

The monitoring and control of TC in water require efficient measurement and degradation. The metal–organic framework has been widely used in environmental remediation owing to satisfactory adsorption and fluorescence performance, especially for those that were functionalized [9,10,11,12]. However, the requirement of devices causes difficulty of on-site tests, and the adsorption efficiency is commonly affected by environmental conditions. To improve the applicability and efficiency, some metal oxides and sulfides possessing peroxidase and photocatalysis features have been developed, such as TiO_2_ and ZnS [13,14]. These materials are capable of achieving a rapid visible color conversion with H_2_O_2_ and 3,3′,5,5′-tetramethyl benzidine (TMB), and further produce a measurable color distinction with the target. Meanwhile, under light conditions, it can produce •O_2_^−^ and •OH, which are able to degrade the pollutants via the oxidation procedure. This approach is efficient, economic, and free of secondary pollution, whereas the catalysis and degradation efficiency of the independent nanomaterials are limited, and the unspecific aggregation additionally inhibits the property.

To improve the functional performance of nanomaterials, nanocomposites are proposed. As the reported information shows, the affinity and degradation rate of TiO_2_/g-C_3_N_4_ were much higher than those of TiO_2_ [15]. Similarly, in comparison to ZnS, the one doped imvite was more efficient in catalysis [16,17]. Admittedly, these improvements are merely from the transformation of the material construction, it requires an ordered and controlled modification, and the effect is limited. Performance complementarity is a novel requirement for desirable nanocomposites [18,19]. Therefore, a heterojunction material was developed. It is a semiconductor material composed of two different elements or materials, such as Au@CdS/WO_3_ and PNCs–COFs [20,21], which were wildly applied in photocatalytic degradation owing to the prevention of back charge transfer and enhanced light absorption. Inspired by this, a p-type semiconductor material such as CuS becomes a potential functional ligand for ZnS, which is the n-type one, and the constructed nanocomposite with the p–n heterojunction suggests a more efficient approach to perform the TC measurement and degradation [22].

Herein, Cu(NO_3_)_2_ was doped into ZnS to prepare a p-CuS/n-ZnS nanocomposite via the cation exchange method at room temperature. Under the condition of the constructed p–n heterojunction, the gap width and electron mobility efficiency of this material were improved, which further enhanced the immanent peroxidase activity and photocatalytic performance. The addition of H_2_O_2_ and TMB could produce plenty of •OH to support a color conversion, while in the presence of TC, the colorimetric signal was further enhanced, and the distinction level was sensitive to the target amount. Additionally, under light conditions, the •O_2_^−^, •OH, and h^+^ produced by photocatalysis were capable of degrading TC (Figure 1). In comparison to other functionalized nanocomposite, this work focuses on the complementarity and configuration features of materials, and aims to achieve a dual application of determination and degradation for the pollutants.

## 2. Materials and Methods

### 2.1. Reagents and Apparatus

The Zn(CH_3_COO)_2_, Cu(NO_3_)_2_·3H_2_O, tetracycline hydrochloride, streptomycin, L-arginine, L-lysine, L-tyrosine, and L-histidine were purchased from Macklin Bio-Chem Technology Co., Ltd. (Shanghai, China). The isopropanol, triethanolamine, ceftriaxone sodium, p-benzoquinone, oxytetracycline, aureomycin, sodium acetate, acetic acid, KCl, MnSO_4_·H_2_O, CaCl_2_, H_2_O_2_, 3,3′,5,5′-tetramethylbenzidine (TMB), thiourea, ethanol, and terephthalic acid were obtained from Aladdin Bio-Chem Technology Co., Ltd. (Shanghai, China). The reagents used in assays were of analytical grade. The ultrapure water was obtained through an HFB-10 purifier (Shanghai, China).

The F-7000 fluorescence spectrophotometer (Hitachi, Japan) was used to perform the catalytic mechanism analysis of the p-CuS/n-ZnS. The UV-3900H ultraviolet spectrophotometry (Hitachi, Japan) was utilized to perform catalytic procedure study and target measurements. For the material characterization, the Ultima IV X-ray Powder Diffractometer (XRD) (Bruker, Germany) was employed to perform surface diffraction investigation, the S-4800 field emission scanning electron microscope (SEM) (Hitachi, Japan) and JEM-2100HR 200 kV field emission transmission electron microscope (TEM) (JEOL, Japan) were used to perform the size and morphology analysis, and the Axis UtraDLD X-ray energy spectrometer (Shimadu, Japan) was utilized to obtain the energy-dispersive spectrum (EDS). The CHI660C electrochemical workstation (Shanghai, China) was employed to conduct the electrochemical performance investigation of the p-CuS/n-ZnS. The PLS-SXE300 xenon lamp (Beijing, China) was used to produce photocatalytic degradation assays.

### 2.2. Synthesis of p-CuS/n-ZnS

First, ZnS was synthesized. An amount of 60 mL of 50% ethanol containing 13 mM thiourea and 6 mM Zn(CH_3_COO)_2_ was uniformly stirred for 50 min, and incubated in a reactor at 160 °C for 12 h. After cooling to room temperature, the product was washed using ethanol and water three times, respectively. The product was then incubated in a vacuum-drying oven at 60 °C for 24 h. An amount of 100 mg of the prepared ZnS was dispersed in 30 mL of ultrapure water that contained 1 mM Cu(NO_3_)_2_·3H_2_O. Subsequently, the mixture was subject to magnetic stirring (700 r/min) at room temperature for 3 h. The product was washed using ethanol and water three times and dried in a vacuum-drying oven at 70 °C for 12 h to obtain the p-CuS/n-ZnS. Through adjusting the level of Cu(NO_3_)_2_·3H_2_O, the nanocomposite with different mass ratios was capable of being prepared.

### 2.3. Determination of Application of p-CuS/n-ZnS

First, the investigation of the peroxidase performance of p-CuS/n-ZnS was performed. An amount of 2 mL of 0.2 M acetate-acetate buffer solution (pH 4.0) contained 0.12 mg/mL of p-CuS/n-ZnS, and 0.8 mM TMB and 0.6 mM H_2_O_2_ were incubated at room temperature for 10 min. The colorimetric signal was measured as an absorbance value by ultraviolet spectrophotometry (λ = 652 nm). In contrast, an equal amount of CuS or ZnS was used to perform the assays.

Subsequently, in the same interaction system, different concentrations of target TC were added to take the assays. The distinctions of the colorimetric signals before and after addition of the TC were recorded to produce a calibration plot. In the practical assays, the water sample taken from the Ganjiang river was stored to avoid light at room temperature for 2 days, and then filtrated by a polyethersulfone (PES) membrane (0.22 µm pore diameter). An amount of 2 mL of the filtrated water was then added into 2 mL of 0.4 M acetate-acetate buffer solution (pH 4.0) containing 0.24 mg/mL of the p-CuS/n-ZnS, 1.6 mM TMB, and 1.2 mM H_2_O_2_ to incubate for 10 min, and the measurements were taken in the same procedure. In the spiked assays, different concentrations of TC were additionally added into the samples before the pre-treatment, and the measurements were made under the same conditions.

### 2.4. Degradation Application of p-CuS/n-ZnS

The optimized amount of the p-CuS/n-ZnS was added into 100 mL of the water sample containing 10 mg/L of TC. The solution was incubated to avoid light for 40 min with continual magnetic stirring (700 r/min). Subsequently, the solution was transferred under the xenon lamp (λ > 420 nm) to perform photocatalytic degradation. In the following period, 3 mL of the water sample was then taken to measure the absorbance value every 20 min. According to the signal conversion trend, the degradation rate of TC was obtained.

## 3. Results and Discussion

### 3.1. Material Characterization

The p-CuS/n-ZnS nanocomposite was synthesized via the cation exchange method. To investigate its composition and structure, a TEM assay was first performed. As Figure 1A shows, the material was spherical with a diameter of 2 µm. Through the high-resolution TEM graph (Figure 1B), two distinct lattice fringes with spacings of 0.301 and 0.312 nm were observed, which belong to CuS (102) and ZnS (002), respectively [23]. Subsequently, the material was characterized by SEM. From Figure 1C, the spheroidal structure of the material was confirmed. However, this structure was distinct to that of ZnS, which was a rodlike structure (Appendix A). The spherical structure was mainly caused by the doping of Cu^2+^ during the synthesis. The added Cu(NO_3_)_2_·3H_2_O produced amounts of dispersed Cu^2+^, which is capable of attacking the prepared ZnS actively to make ion exchanges [24,25]. The smashed ZnS was then employed as the core substrate to load the Cu^2+^, and finally formed a spherical nanoparticle. The followed EDS analysis with spot (Figure 1D) and map (Appendix A) scanning both illustrated the material containing the elements of Cu, Zn, and S. This preliminary confirmed the synthesis of p-CuS/n-ZnS.

To further analyze the element bonding state of the prepared nanocomposite, an XPS assay was performed. As Figure 2A shows, the spectrum of the p-CuS/n-ZnS was similar to that of ZnS in addition to a peak of Cu element. Admittedly, the spectrum additionally contained the peaks of C and O elements. This may be caused by the O_2_ and CO_2_ from the air. The high-resolution XPS spectrum (Figure 2B) showed two specific peaks at 1021.99 and 1045.03 eV in the ZnS spectrum, which belong to Zn 2p_3/2_ and Zn 2p_1/2_., respectively. For the p-CuS/n-ZnS spectrum, the peaks were presented at 1022.07 and 1045.25 eV. This conversion may be caused by the low crystallinity of the nanocomposite. In addition, Figure 2C displays two specific peaks at 161.93 and 163.30 eV in the ZnS spectrum, which belong to S 2p_3/2_ and S 2p_1/2_, respectively, whereas, in the p-CuS/n-ZnS spectrum, they transferred to 161.53 and 162.47 eV, respectively. Meanwhile, another peak was observed at 169.14 eV. This was recognized as S^4+^, produced by the simultaneous coordination of the S to the Cu and Zn. Finally, Figure 2D exhibits two specific peaks at 932.13 and 952.12 eV, which belong to Cu 2p_3/2_ and Cu 2p_1/2_, respectively These results solidly support the synthetic product being p-CuS/n-ZnS. From the detailed XPS data, the atomic concentration of Cu was 17.14%, whereas that of Zn was 34.59%. As for the mass concentration, the percentage of Cu was 22.24% and that of Zn was 46.17%.

### 3.2. Performance Investigation

To investigate the influence of the doped Cu^2+^ amounts on the p-CuS/n-ZnS performance, a series of comparison assays were performed. First, the nanocomposites with CuS mass ratios of 10%, 15%, 20%, 25%, and 30% were prepared. The XRD test results (Figure 3A) showed that the increase in CuS level led to a continuous broadening of the diffraction peak at 28.89° (002), and the peak intensity gradually declined. This indicates that the lattice plane of ZnS was weakened, and the Cu^2+^ ceaselessly loaded onto the ZnS to generate CuS. Meanwhile, the regular variations in the crystalline form feature suggest that this nanocomposite was formed via the bridging effect of the inherent p–n heterostructure rather than physical accumulation.

The specific surface area of the nanocomposites with different levels of CuS were additionally investigated. As the results of the N_2_ adsorption–desorption isothermal curve assays illustrate in Figure 3B, the nanocomposite containing 25% of CuS exhibited the optimal value (13.16 m^2^/g), which was threefold that of the pure ZnS (4.12 m^2^/g), suggesting that it exhibits the largest specific surface area. Meanwhile, both linear shapes of the ZnS and the nanocomposites belonged to the IV type isotherms and presented H3 (P/P0 > 0.8)-type hysteresis loops, indicating that the materials exhibited an aperture structure. According to the aperture distribution curve (Figure 3C), the nanocomposite containing 25% of CuS showed the optimal state, the average pore volume was 0.057 cm^3^/g, which was fivefold that of the ZnS, and the pore width mean was 15.01 nm. It is interesting that the specific surface area and aperture were reduced once the CuS amount achieved 30%. This is because the aggregation effect of the CuS caused an accumulation of the nanocomposites, and the limited dispersal influenced the catalytic performance of the material.

The specific surface area is commonly related to catalytic property of material. Thus, the peroxidase activity of these p-CuS/n-ZnS nanocomposites was checked. In the solution containing H_2_O_2_ and TMB, 6 mg/mL of ZnS and the nanocomposites were added to take colorimetric assays by ultraviolet spectrophotometry. As the results (Figure 3D) show, owing to the cooperative effect, the produced color conversion of the nanocomposites was much higher than that of the independent ZnS, and the most obvious is that containing 25% of CuS. To further assess the peroxidase activity, the Michaelis–Menten dynamic model was introduced to calculate the constant value (Km). The results (Appendix A) showed that the Km values for H_2_O_2_ and TMB were 3.53 and 0.39 mM, which is better than the performance of horseradish peroxidase.

Subsequently, the photocatalytic performance of the prepared p-CuS/n-ZnS was investigated, which aims to assess the electron transfer efficiency and photogenerated electrons property. The electrochemical impedance spectrum (Appendix A) illustrates that the Nyquist curve radius of ZnS was largest, indicating the least electron transfer efficiency. With the addition of CuS to the 25% level, it displayed the smallest radius, suggesting that the electron transport resistance was minimum, and the catalytic efficiency was highest. To prove this standpoint, a steady-state fluorescence test was performed. The results (Appendix A) showed that the fluorescence intensity of the nanocomposite with 25% of CuS was lowest, explaining that the recombination rate of photogenerated electrons and material holes was limited, which led to the highest photocatalytic activity [26,27,28]. In addition, the transient photocurrents of the materials were measured to further verify the reliability of the obtained results. As Appendix A shows, the photocurrents intensity of the nanocomposites was continuously raised until the level of CuS achieved 25%, suggesting that the contained photogenerated electron holes were the most under this status. However, when the CuS amount reached 30%, the photocurrent signal declined. This was caused by the excess CuS blocking light absorption. The p-CuS/n-ZnS containing 25% CuS was proved to possess satisfactory enzymatic and photocatalytic performance, and this construction was confirmed for further use.

### 3.3. Colorimetric Determination of TC

Encouraged by the favorable performance of the p-CuS/n-ZnS, it was then employed to perform a colorimetric detection for TC. First, the detection conditions including the dosage, TMB amount, pH, and temperature were optimized. From Appendix A, it displays the strongest absorbance signal when the amount of the p-CuS/n-ZnS achieved 0.12 mg/mL. Less or more p-CuS/n-ZnS amount can lead to an insufficient reaction or agglomeration with the substrate, and prevent the catalytic procedure. Similarly, the optimal TMB concentration was confirmed as 0.8 mM (Appendix A). Subsequently, these amounts were further used to investigate the effect of pH value and incubation temperature. As the result shows in Appendix A, the optimal absorbance signals were presented at pH 4.0 and 25 °C.

After optimizing the detection condition, the colorimetric determination of TC was performed. The blank test displayed a colorless signal (inset a, Figure 4A), and a light blue signal was produced by p-CuS/n-ZnS catalyzing H_2_O_2_ and •OH oxidizing TMB (inset b, Figure 4A). Following the addition of TC, the color became gradually deeper with the concentration increase (inset c~h, Figure 4A), indicating that the TC can promote catalytic interaction. Subsequently, the TCs with more different concentrations were utilized to perform the UV tests. By observing the absorbance signals, and deducting the background signal merely caused by the p-CuS/n-ZnS catalysis, a signal conversion was obtained (Figure 4A), which further obtained a linearity in the concentration range of 0.25~3.0 µM (Figure 4B). The equation was ∆A = 0.051CTC + 0.315 and the regression coefficient was 0.998. According to the linearity, the limit of detection (LOD) was calculated as 20.94 nM (3σ/S). In comparison with the other reported determination method for TC, this strategy possesses certain benefits in the linear range and LOD (Appendix A).

To verify the mechanism of the colorimetric detection, terephthalic acid (TA), which can interact with •OH to produce a fluorescence signal, was introduced in the further assays. As the results show in Figure 4C, when the H_2_O_2_ occurred with TA, a low fluorescence signal was obtained owing to the decomposition of H_2_O_2_ producing few •OH (line e). After the addition of TC, the fluorescence signal was slightly increased (line f), suggesting that the TC can promote the decomposition of H_2_O_2_. In contrast, the fluorescence signal produced by p-CuS/n-ZnS catalyzing H_2_O_2_ was more obvious (line g), and this signal was further raised dramatically with TC (line h). These demonstrated the preconceived mechanism of catalytic interaction.

The selectivity of the determination strategy was then assessed. Some common antibiotics and ions including streptomycin, ceftriaxone sodium, oxytetracycline, chlortetracycline, L-arginine (L-aRg), L-lysine (L-Lys), L-tyrosine (L-Tyr), L-histidine (L-HIS), K^+^, Mn^2+^, and Ca^2+^ were utilized as interference to produce the assays. The results (Figure 4D) showed that the signal response to TC (blue bar) was much higher than the others. In addition, the signals of oxytetracycline and chlortetracycline were also obvious. This is because they exhibited a similar structure to that of TC. However, compared with TC, their responses were limited.

To further assess the feasibility for practical application, this strategy was used to detect the TC in the water sample taken from the Ganjiang river. After filtering by the membrane with 0.22 µm diameter, the water was subjected to assay. The water was detected as a negative sample, and this was proved by the contrast method using HPLC [29] (Table 1). In the following spiked assays, the TC standards were added into the water sample before pretreatment, and the detection results illustrated that the recoveries were 106% and 99% with RSDs of 2.54% and 1.22%, respectively, suggesting a satisfactory practicability.

### 3.4. Photocatalytic Degradation of TC

The condition of photocatalytic degradation was optimized. First, 0.5 g/L of p-CuS/n-ZnS and 10 mg/L of TC were mixed into 100 mL of acetic acid-sodium acetate buffer with different pH values. The conversions of the adsorption equilibrium concentrations illustrated that the adsorption amount of TC at pH 5.0 was optimal, which further resulted in the highest degradation efficiency (Figure 5A). Subsequently, the appropriate use dosage of the p-CuS/n-ZnS was investigated. The results (Figure 5B) showed that 0.5 g/L was its optimal usage. Under the optimal condition, p-CuS/n-ZnS was utilized to degrade 10 mg/L of TC. The results (Appendix A) showed that the removal included two stages. The first adsorption stage (−40~0 min) removed 65% of TC. After the further photocatalytic stage (0~120 min), the remnant of TC was ~0.9 mg/L, indicating that the total degradation rate arrived at 91%. The calculation plot relating to the apparent rate demonstrated that the reaction conformed to the rule of the pseudo-first-order reaction [30,31], and the constant value (*K*) was 0.0119 min^−1^ (Appendix A).

The photocatalytic degradation relies on the photoexcited electrons and holes in the conduction band (CB) and valence band (VB). These values were investigated via UV-Vis DRS assays. As the results show in Appendix A, the *E*g can be assessed by the Kubelka–Munk equation:α*hv* = A(*hv* − *E*g)^n/2^
where α, *h*, A, and *E*g are absorptivity, light frequency, Planck constant, and bandgap value, respectively. As a result, the *E*g of CuS and ZnS were 1.45 and 3.31 eV, and that of the p-CuS/n-ZnS was 2.58 eV. Subsequently, their *E*_CB_ and *E*_VB_ were calculated by the equation as follows:*E*_CB_ = X − *E*e − 0.5 *E*g
*E*_VB_ = *E*_CB_ + *E*g
where X is the electronegativity of the semiconductor and *E*e is ~4.5 eV. Therefore, the *E*_CB_ of CuS and ZnS are 0.04 and −0.90 eV, and their *E*_VB_ are 1.50 and 2.42 eV, respectively. Additionally, the *E*_CB_ and *E*_VB_ of the p-CuS/n-ZnS are −0.61 and 2.05 eV, respectively.

The p-CuS/n-ZnS nanocomposite enhanced photocatalytic efficiency mainly due to the heterogeneous structure, which was the wide bandgap of the n-type combined with the narrow bandgap of the p-type, and promoted the separation of photoexcited electrons from hole pairs [32]. Meanwhile, the addition of CuS to ZnS led to a larger specific surface area and more active sites. The photocatalytic degradation mechanism of p-CuS/n-ZnS to TC is shown in Figure 6A, the procedure is deduced as followed:CuS/ZnS + *hv* → CuS/ZnS (h^+^) + CuS/ZnS (e^−^)e^−^ + O_2_ → •O_2_^−^H_2_O + h^+^ → •OH + H^+^TC + •O^2−^ + •OH + h^+^ → degradation

Under light conditions, the electrons generated by p-CuS/n-ZnS were combined with O_2_ in the air to produce •O_2_^−^. In addition, the photogenerated holes in VB of the p-CuS/n-ZnS can react with H_2_O to produce more •OH. Undergoing the cooperation effects of the produced •O_2_^−^, •OH, and h^+^, the target TC was degraded.

To further understand the effect level of the •O_2_^−^, •OH, and H^+^ to TC degradation, a comparison assay was performed using 10 mg/L of triethanolamine (TEOA), isopropanol (IPA), and 1,4-benzoquinone (BQ), respectively. The results (Figure 6B) showed that the degradation rate decreased by 6% and 11% after the addition of TEOA and IPA, respectively, suggesting that the effects of H^+^ and •OH were limited, whereas the presence of BQ led to a 22% decline, indicating that the •O_2_^−^ resulted in a major effect in degradation. To investigate the repeatability of the p-CuS/n-ZnS, the result was collected after one degradation assay for the next one. The results (Figure 6C) showed that the photocatalytic degradation rate of 0.5 g/L of p-CuS/n-ZnS to 10 mg/L of TC remained above 86% after four cyclical degradation procedures. Meanwhile, after assessing the structure and morphology of the p-CuS/n-ZnS before and after performing photocatalytic assays by XRD (Figure 6D) and SEM (Appendix A), similar characterization results demonstrated a high stability of this prepared nanocomposite.

## 4. Conclusions

A nanocomposite of p-CuS/n-ZnS was prepared by the cation exchange method. The p–n heterojunction resulted in a threefold improvement in the catalytic property. Meanwhile, the nanocomposite contained dual performances of mimicking the enzyme and photocatalysis, which was employed to perform the colorimetric determination and degradation of TC. Undergoing the induced effect of p-CuS/n-ZnS oxidizing H_2_O_2_, the presence of TC caused a promotion of the TMB color conversion. The signal responded within 10 min, and the determined sensitivity achieved 20.94 nM. In addition, the degradation rate for TC was 91% within 120 min. However, the selectivity of the p-CuS/n-ZnS-based determination method was limited, whereas this issue could be resolved via a further separation procedure or adding a specific masking agent. In comparison to other nanocomposites, the proposed p-CuS/n-ZnS was constructed based on the complementary structural features, and it realized multiple-method uses, providing prospect support for the monitoring and control of antibiotics.

## Data Availability

Not applicable.

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
