# Peer review of "Dual Application: p-CuS/n-ZnS Nanocomposite Construction for High-Efficiency Colorimetric Determination and Photocatalytic Degradation of Tetracycline in Water"

_nanomaterials, 2022, doi:10.3390/nano12234123_

Round 1

Reviewer 1 Report

This manuscript reported “Dual application: p-CuS/n-ZnS nanocomposite construction for high efficient colorimetric determination and photocatalytic degradation of tetracycline in water”. This proposed work exhibited a good value for the detection and degradation of tetracycline capacity. Thus, author should address as follows my comments in order to improve the quality of the manuscript. My recommendation is to reconsider after the Major revisions.

My comments:

1.     Abstract requires more technical achievements from the proposed work to highlight the novelty of the work.

2.     Keywords should not identical to title words, revise it.

3.     It should reconstruct the content abstract and the length of the introduction part. Authors should maintain a single frame of language understanding and phrase maintenance throughout the paper.

4.     The authors should clearly explain the innovation, research gap, market gap, market demand, and importance of their work in the manuscript's introduction. They should justify the value of the work and compare their work with previously similar published papers. They should develop the photocatalysis advantage and applications compared to other known systems. The introduction section needs to be elaborated.

5.     In Fig. 3A, Author should give appropriate references with JCPDS card number for both CuS and ZnS.

6.     Author should explain, why Figure S1 rod shape in the SEM image. Except Fig. S1 other TEM and SEM images didn’t shows the rod shape.

7.     Previously reported work showing good sensing and photocatalytic properties for tetracycline than this proposed work. Author should demonstrate novelty of this work.

https://doi.org/10.1007/s00449-009-0371-4

https://doi.org/10.1080/00032719.2012.670784

https://doi.org/10.1016/j.chemosphere.2013.02.066

https://doi.org/10.1016/j.jcis.2018.02.067

8.     Should compare detection limit, linear ranges and sensitivity with previously reported tetracycline sensor.

9.     Author should perform FT-IR analysis and discuss in the revised version.

10.  Removal percentage and qe(mg/g), which to depend on to qualify adsorbent quality? Authors should plot both qe(mg/g) and removal(%).

11.  It is required that the author demonstrate the flaws and potential improvements in the conclusion.

12.  The grammar of the manuscript should be polished; the typos and grammatical errors are scattered throughout the paper and need to be corrected with utmost care.

Author Response

The response for Reviwer 1 is in attachment.

Reviewer 2 Report

Tu and coworkers reported a very interesting work preparing a p-CuS/n-ZnS nanocomposites for photocatalytic degradation of tetracycline in water. The work is well performed and organized. However, there are still some flaws in the manuscript, including experiments, writing and references. Therefore, a major revision is needed.

1. The morphology of the materials is quite confusing. From scheme 1, seems like the CuS particles is on the ZnS nanoparticles. However, from Figure 1a and 1c, I cannot see the same morphology. Why?

2. Figure 1c, there are large spheroidal structure, which I assume is ZnS. And there are small particles that might be CuS. But those big particles and small particles are separated. Can they still form a heterojunction?

3. Can you also use XPS or EDS to confirm the Cu ratio inside the nanocomposites?

4.  The advantages of heterojunction nanocomposite photocatalysts should be further introduced, such as preventing the back charge transfer and enhanced light absorption. Some recent important examples of heterojunction photocatalysis (https://doi.org/10.1021/acsmaterialslett.1c00785; https://doi.org/10.1039/C5CC05211D)must be cited.

5. Figure 5, please clarify what is the region from -40 mins to 0 mins.

6. PH =5 .0 seems like the best condition. Any hypothesis for why is this PH is beneficial for the photocatalytic performance?

7. Figure 6a, how do authors know the CB and VB of the p-CuS/n-ZnS nanocomposites?

Author Response

The response to Reviewer 2 is on the attachment.

Round 2

Reviewer 1 Report

Authors have performed the revision in perfect manner. So, I strongly recommend for publication.

Reviewer 2 Report

The revised version is in very good shape. I suggest publishing this excellent work as it is